# Free Myocutaneous Flap Assessment in a Rat Model: Verification of a Wireless Bioelectrical Impedance Assessment (BIA) System for Vascular Compromise Following Microsurgery

**DOI:** 10.3390/jpm11050373

**Published:** 2021-05-04

**Authors:** Yao-Kuang Huang, Min Yi Wong, Chi-Rung Wu, Yung-Ze Cheng, Bor-Shyh Lin

**Affiliations:** 1Division of Thoracic and Cardiovascular Surgery, Chia Yi Chang Gung Memorial Hospital, College of Medicine, Chia-Yi and Chang Gung University, Taoyuan 33302, Taiwan; huang137@icloud.com (Y.-K.H.); mynyy001@gmail.com (M.Y.W.); 2Institute of Imaging and Biomedical Photonics, National Yang Ming Chiao Tung University, Tainan 71150, Taiwan; luck511286@gmail.com; 3Department of Medical Research, Chi-Mei Medical Center, Tainan 71004, Taiwan; asaliea.cheng@gmail.com

**Keywords:** flap grafting, microsurgery, biosensor, bioelectrical impedance, rat

## Abstract

Background: Microvascular tissue transfer is a common reconstructive procedure. We designed a bioelectrical impedance assessment (BIA) system for quantitative analysis of tissue status. This study attempts to verify it through the animal model. Methods: The flaps of the rat model were monitored by the BIA system. Results: The BIA variation of the free flap in the rat after the vascular compromise was recorded. The non-vascular ligation limbs of the same rat served as a control group. The bio-impedance in the experimental group was larger than the control group. The bio-impedances of both the **thigh**/**feet** flaps in the experimental group were increased over time. In the **thigh**, the difference in bio-impedance from the control group was first detected at 10 kHz at the 3rd and last at 1 kHz at the 6th h, after vascular compromise. The same finding was observed in the **feet**. Compared with the control group, the bio-impedance ratio (1 kHz/20 kHz) of the experimental group decreased with time, while their variation tendencies in the thigh and feet were similar. Conclusions: The flap may be monitored by the BIA for vascular status.

## 1. Introduction

Free tissue transfer by microsurgery is a commonly performed procedure within reconstructive surgery [1,2,3]. Using the autograft method, microvascular free tissue transfer can effectively rescue lethal surgical complications [3,4]. Monitoring of perfusion status, detection of thrombus formation within the capillaries after microsurgery and timely rescue of the flap are the crucial elements that contribute to a perfect free skin flap surgery. Physicians often observe variations in the flap edge, flap color and flap flexibility and use their previous experience to evaluate the state of the flap [5,6]. However, there are no objective parameters to help the physician make a correct judgement.

Measuring the surface temperature of grafted skin flaps is the simplest way of monitoring skin flap grafting. Physicians used skin temperature indicators to conduct free flap monitoring [7]. When the center location of the skin dropped by more than 3 °C from the baseline, it might have encountered arterial thrombosis. However, the sensitivity for monitoring the surface temperature of skin flap grafts is not high enough [8]. Several pieces of equipment, such as the Doppler ultrasound, a microdialysis system, a tissue pH monitoring system, laser Doppler flowmetry and color Duplex sonography, have been previously used to evaluate skin flaps [7,8,9]. The Doppler ultrasound device and the color Duplex sonograph can monitor blood flow within the transferred flaps to indirectly evaluate their condition. The microdialysis system is based on a perfusion sampling and dialysis technique that estimates free transverse rectus abdominis myocutaneous (TRAM) flaps [9]. Partial ischemia in free TRAM flaps could be indirectly estimated through variations in glucose, glycerol and lactate concentrations. The tissue pH monitoring system indirectly estimates the failure of free flaps and vascular complications using variations in pH that are caused by blockages in the artery. Laser Doppler flowmetry and tissue spectrophotometry use blood flow parameters and oxygen saturation to distinguish between blocked arteries and venous congestion [10]. Although the above methods could help the physician evaluate the condition of flaps, most of them are invasive and expensive and require the operator to be experienced.

Bioelectrical impedance analysis (BIA) has been developed to evaluate changes in human tissue composition and animal experiments [11]. The basic concept of BIA is to inject a current with different frequencies into the tissue to estimate the whole bio-impedance of the human tissue, which is contributed to by the bio-impedances of the different human tissue components [12,13]. It is a non-invasive measurement and can be used to monitor changes in human tissue composition in real-time.

The advantages of BIA are that it is non-invasive and provides real-time measurements, and it could have the potential as a helpful tool for evaluating the status of tissue flaps after microsurgery. In the current study, physiological parameters were monitored using the BIA technique to measure changes in bio-impedance and to verify the effectiveness of BIA in a rat model.

## 2. Materials and Methods

### 2.1. Bioelectrical Impedance Analysis

When an electrical current passes through different types of human cells or tissues (fluids, adipose, muscles, etc.), these tissues have different levels of conductivity due to the different numbers of electrolytes in them. In general, adipose tissue and bone have poorer conductivity, whereas fluids and muscles provide better conductivity. Lower frequency electrical currents have difficultly penetrating the cell membrane when they pass through a biological cell (i.e., most of the electrical current passes through the extracellular cell). Higher frequency bio-impedance may give more information on the intracellular and extracellular fluid [14]. Multiple-frequency BIA can be easily classified into several different frequency bands: the alpha domain (1~10 kHz), the beta domain (10~50 kHz) and the gamma domain (>100 kHz) [15]. Bio-impedance in the alpha domain may be associated with information on tissue interfaces, while the beta domain is associated with the polarization of cellular membranes, proteins and other organic macromolecules. Bio-impedance in the gamma domain may be associated with the polarization of water molecules [16]. Bio-impedance in the alpha and beta domains is most frequently used because the differences between normal and pathological tissues can be observed in their variations. Therefore, the change in the health of the flap may be effectively observed by the bio-impedance measurement in the alpha and beta domains.

Bio-impedance systems typically use a voltage-controlled current source, such as a Howland current source, to inject a known current into the tissue primarily for safety purposes but also to eliminate the need to measure the injected current into the tissue. In addition, safety limits according to international standards have been set with regards to currents into tissues: 100 μA from 0.1 Hz to 1 kHz, 100f μA from 1 to 100 kHz, and 10 mA for frequencies greater than 100 kHz, where f is the frequency in kHz. A 5 μA can be beneficial in terms of power consumption but maybe not that great for the signal-to-noise ratio. The self-assembled bio-impedance parameter monitoring device used in the current study is shown in Figure 1a. The bio-impedance parameter monitoring device was designed to measure tissue bio-impedance signals. The back-end host system was designed to analyze the raw data and to estimate, display and store the bio-impedance parameters. To ensure no damage or pain was caused, the current passing through the biological tissue was <5 μA. The excitation buffer can provide 2.5 Vp-p steady excitation voltage, and the reference resistance is set to about 5 M ohm. Therefore, the injected current will be limited below 5 uA. The block diagram for the bio-impedance parameter monitoring device is shown in Figure 1b. It mainly contains a microprocessor, a wireless transmission circuit, a steady voltage source circuit, a voltage divider circuit, a voltage acquisition circuit and a probe. The cost of this system is 200 USD.

For functional bio-impedance analysis, the steady voltage source circuit was designed to provide a steady voltage with a specific frequency ranging from 1 to 20 kHz, which can be controlled by the microprocessor. The cut-off frequency of the low-pass filter in the steady voltage source circuit was set to 25 kHz. The generated steady voltage will then pass through the voltage divider circuit and the stainless-steel electrodes. The pair of stainless-steel electrodes will be placed on the region of tissue being measured. When generated, the steady voltage passes through the tissue via the electrodes and will then be attenuated due to tissue bio-impedance. The voltage acquisition circuit then receives the attenuated steady voltage signal and estimates the bio-impedance of the tissue. The gain of the voltage acquisition circuit is adjustable and can be set to 1, 2, 5 or 10. The sampling rate of the analog-to-digital converter in the microprocessor is set to 200 kHz. Under the consideration of simple operation and implementation of the designed device, bipolar measurement is used in this study. In bipolar measurement, the electrode interface impedance may cause the contributor in the measured impedance. Under the same measurement condition, the relative impedance change caused by the change in the health of the flap was monitored. Here, the distance between the two electrodes is set to 10 mm. Because the thickness of the flaps is about 1–3 mm, the measuring depth of the designed device should be sufficient. Here, the shape of the used electrodes is a round tip needle, and it can reduce the risk of injuring the flaps. The used electrodes are made of stainless steel can improve the issue of metal oxidation.

### 2.2. Animal Experiment Design and Procedure

In the animal experiment, adult male Sprague Dawley albino rats with a weight of 325 ± 25 g were used. The experimental procedure was approved by the Animal Care and Use Committee of Chi-Mei Medical Center to minimize the discomfort to the animals during the study period. All rats were kept in a temperature-controlled air-conditioned room (23 ± 2 °C) for at least 7 days before this experiment began and were maintained in a 12-h light/dark cycle. After the experiment, all rats were sacrificed using urethane. During the experiment, all rats were first anesthetized by intraperitoneal injection with ketamine, and their hair was trimmed using a razor blade. Then, the right thigh of the animal was surgically dissected, and its muscles, nerves and skin tissue were all divided; their vascular pedicle was isolated but not transected. The left thigh of the animal that did not undergo surgery was still able to supply blood and nutrition from the femoral vessels and was used as the control group (Figure 2). Next, the skin on the right thighs was sutured, and probes from the BIA system were attached to both limbs. After the BIA system started to record, the right vascular pedicles were ligated. The BIA system probes were placed on the thighs of the animal to acquire the bio-impedance every hour using a self-assembling holder to stabilize the probe on the target surface. The total length of the experimental procedure was >8 h.

Six same reparative manipulations were done in rats. Power analysis is listed in Table 1. One-way analysis of variance (ANOVA) was used to analyze the experimental results, and *p* < 0.05 was considered to indicate a significant difference.

## 3. Results

The bio-impedance variation of the skin flap after surgery was investigated. The bio-impedances of both the thigh and feet skin flaps in the experimental group were clearly increased over time. Figure 3 and Figure 4 show the time courses of the average bio-impedance in the thigh and feet skin flaps for the different frequencies. The average bio-impedances were obtained from six separate experimental trials. The standard deviation of the bio-impedance in the experimental group was significantly larger than the control group. In the thigh, the difference in bio-impedance from the control group was first detected at 10 kHz (Figure 3c) at the 3rd h and last detected at 1 kHz (Figure 3a) at the 6th h, after vascular compromise. The same finding was observed in the feet. (earliest in 10 Hz at the 3rd h and latest in 1 Hz at the 6th h). Moreover, with the increase in time, the difference between the bio-impedances of the thigh and feet flaps for the control and experimental groups also significantly increased. Figure 5 shows the average bio-impedance ratio (1/20 kHz). Compared with the control group, the bio-impedance ratio of the experimental group decreased with time, while their variation tendencies in the thigh and feet were similar.

## 4. Discussion

Free tissue transfer by microsurgery is a meticulous procedure for reconstructive surgery following cancer or trauma. The first 48 h after surgery are critical for identifying and salvaging a failing flap. Most clinical teams can only provide routine spot checks on patients with the overwhelming workload. Therefore, continuous monitoring can reduce the flap failing rate. Some devices have been developed for continuous monitoring of tissue flaps after surgery, guts perfusion and myocardium by measuring microvascular parameters [11,13,17,18].

Our study demonstrated that after free flap ischemia after vascular compromise, the bio-impedances of the experimental group gradually increased over time. Metabolically active tissues can encounter a lack of oxygen and nutrition, which can cause severe ischemic injuries [19]. When ischemic injuries exceed the tolerance level of flap cells, the flaps will be destroyed [20,21]. Previous studies have indicated that hypoxia-ischemia directly affects the change in tissue bio-impedance and causes an increase in tissue bio-impedance [22,23,24].

The bio-impedance decreased as the frequency increased. This is because higher frequency currents have a better penetrating ability in tissue, as shown in Figure 3 and Figure 4. When the frequency of the electrical current increases, it penetrates not only the extracellular fluid but also the cell membrane and intracellular fluid [25,26]. At 1 and 5 kHz bio-impedance, the experimental group was significantly different from the control group for 5 h after surgery. At 10 and 20 kHz bio-impedance, the bio-impedance of the experimental group was significantly different from the control group for 3 h after surgery. This also indicates that higher frequency bio-impedance (at about beta domain) may be more sensitive to tissue changes after perfusion compromise of skin flaps grafts. In Figure 5, the average bio-impedance ratio (1/20 kHz) decreased with time after surgical occlusion of the vessels. When ischemic injuries occur, extracellular fluid flowing into intracellular fluid results in an increase in capacitive reactance and resistance [13,18]. However, high-frequency currents have better penetration and can provide more sensitive results associated with skin flap necrosis in deeper tissue. Therefore, compared with the control group, the average bio-impedance ratio of the experimental group decreased with time, and the differences between the groups were higher in the thigh (proximal part of the flap near the vessel stalk) than in feet (Figure 5).

Several techniques, such as pH monitoring, microdialysis and laser Doppler flowmetry, might also be used to enhance survival following skin flap grafting. In 1996, WJ Issing et al. used a pH monitoring system to evaluate skin flap viability [27]. The pH monitoring system, which can be used to estimate the degree of ischemia, has to plant a pH microelectrode into the tissue, and then the pH value will change as the biological tissue becomes ischemic [28]. The pH monitoring system has low complexity, is not expensive and provides real-time data. However, the device is invasive (probe insertion into the flap). The microdialysis system can also be used to analyze the ratio of lactate to pyruvic acid in the cell substance to determine the degree of hypoxia-ischemia and is thus a useful tool for monitoring the viability of free flaps [29,30]. In 2016, L Liasis et al. also used the microdialysis system to evaluate the ischemic condition of limbs in patients with diabetes mellitus after amputation [31]. Although the microdialysis system is accurate, it is expensive and invasive and is not suitable for long-term free flap monitoring. Moreover, the microdialysis system requires an experienced operator to conduct the monitoring. Laser Doppler flowmetry is also used in the clinic to estimate the blood flow value. From the variation in blood flow rate, the physician can indirectly evaluate the state of the grafted skin flap [8]. This method is non-invasive but is highly dependent on the experience of the operator. Our BIA system is non-invasive and gives a fast response for detecting the state of the grafted flap; it was easy to use in this rat model. It is potentially useful in a clinical setting after microsurgery with free flaps. However, the change in this BIA after vascular complication is a minimum of 3 h; thus, the system has no sufficient clinical impact yet. Combination with the other bio-signals, for example, near-infrared spectrum, may increase its clinical usability further.

## 5. Conclusions

This BIA system was validated in this animal experiment for its usefulness in monitoring the vascularity and tissue status of grafted flaps. In the experimental group (transected femoral vessels in the groin to stop the blood supply to the flaps), the bio-impedance values of the flaps were significantly increased over time. Higher frequency electrical currents can better penetrate the tissue and are more stable. The 10 kHz detected the bio-impedance difference earliest (3rd h), and the 1 kHz detected the latest bio-impedance difference between the experimental group and the control group after vascular compromise.

## Figures and Tables

**Figure 1 jpm-11-00373-f001:**
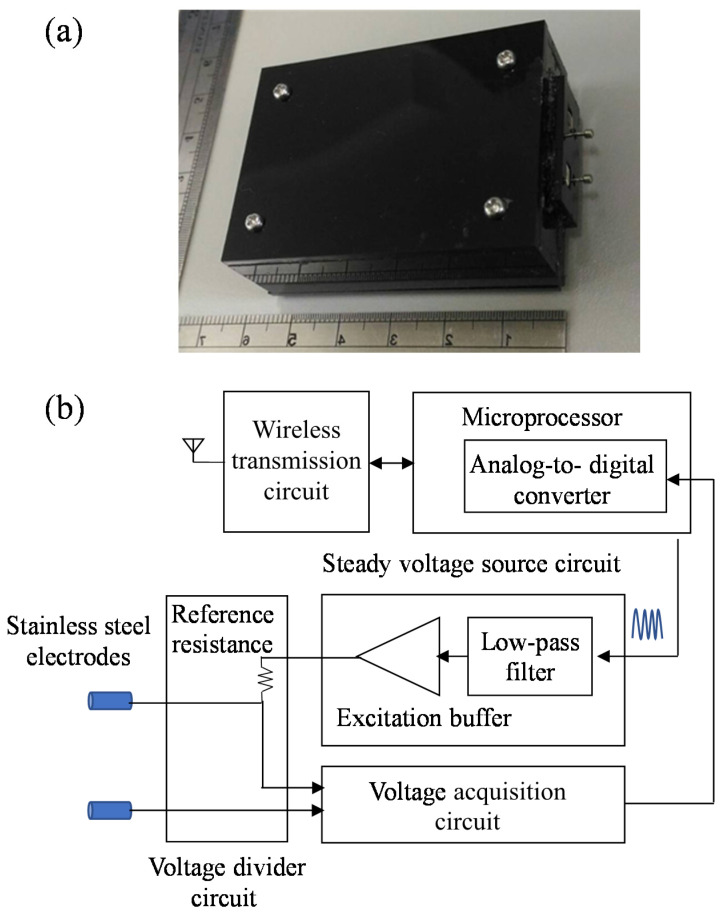
(**a**) Photograph and (**b**) block diagram of the self-assembled bio-impedance parameter monitoring device.

**Figure 2 jpm-11-00373-f002:**
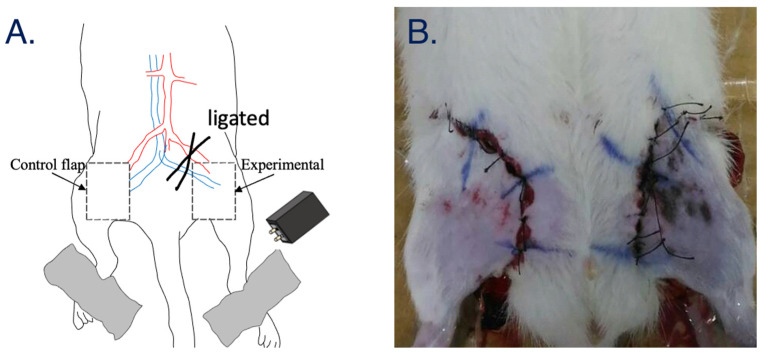
(**A**) A visualized experimental protocol. (**B**) Skin repair before measurement.

**Figure 3 jpm-11-00373-f003:**
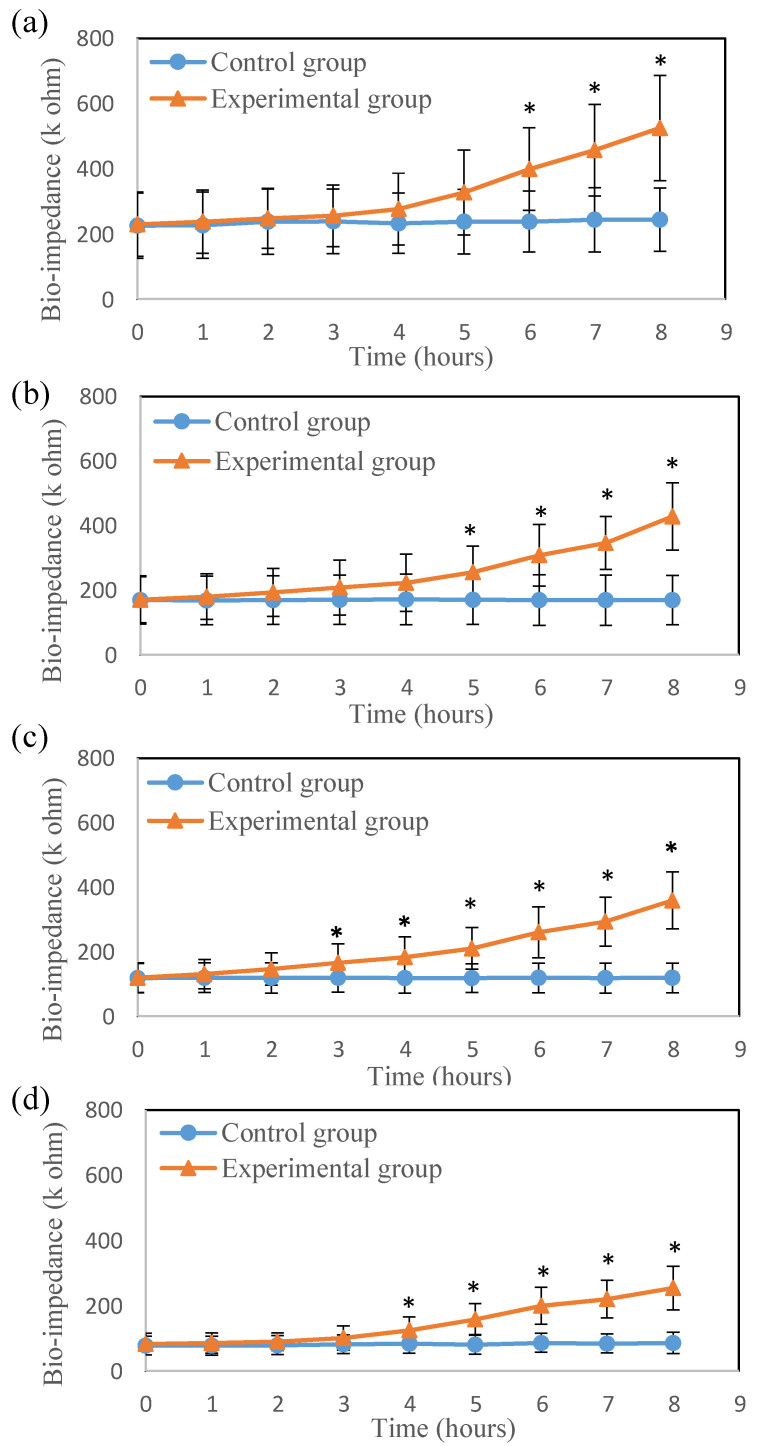
The time course of (**a**) 1 kHz, (**b**) 5 kHz, (**c**) 10 kHz and (**d**) 20 kHz average bio-impedance in the **thigh** flaps (* *p* < 0.05).

**Figure 4 jpm-11-00373-f004:**
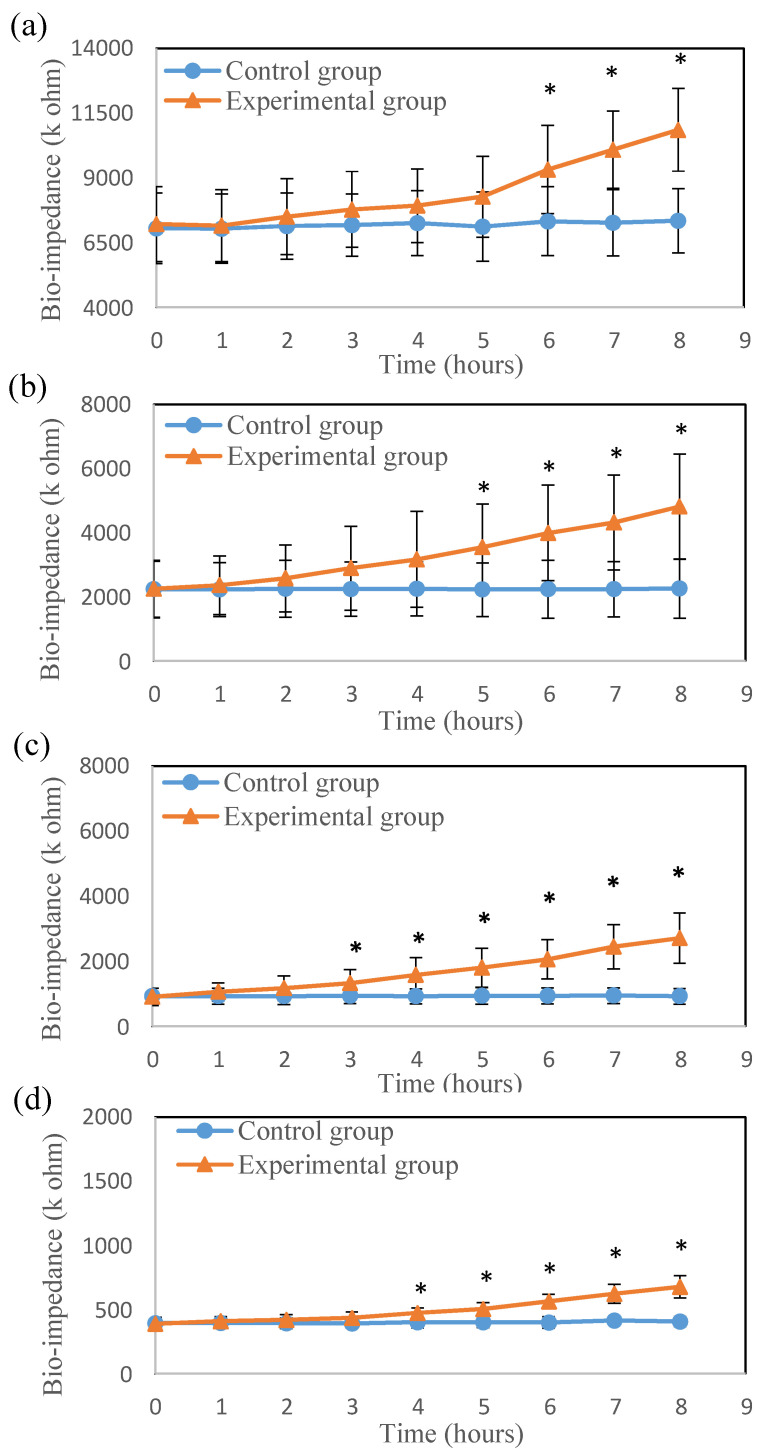
The time course of (**a**) 1 kHz, (**b**) 5 kHz, (**c**) 10 kHz and (**d**) 20 kHz average bio-impedance in the **feet** flaps (* *p* < 0.05).

**Figure 5 jpm-11-00373-f005:**
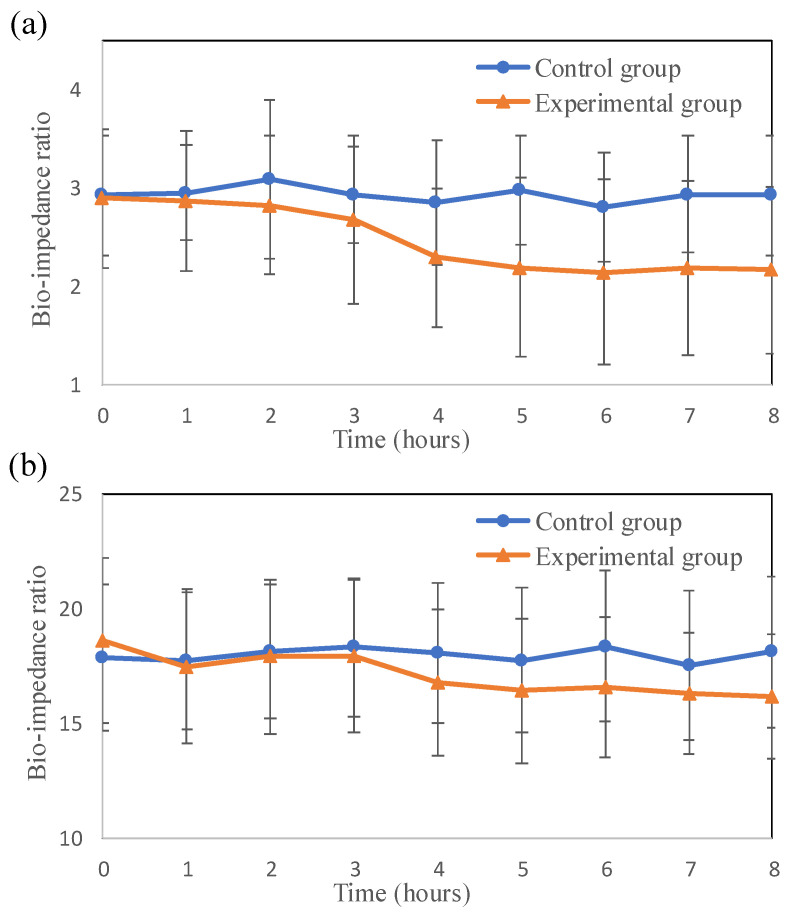
(**a**) Time course of the average bio-impedance ratio (1 kHz/20 kHz) in the **thigh**. (**b**) The time course of the average bio-impedance ratio (1 kHz/20 kHz) in the **feet**.

**Table 1 jpm-11-00373-t001:** Post hoc power analysis for comparing two group means.

Group	Impact Region	1 kHz	5 kHz	10 kHz	20 kHz
Experimental group	Thigh	28.60%	50.90%	81.40%	45.40%
	Feet	31.40%	40.50%	91.60%	3.30%
Control group					

Post hoc was estimated using the means and standard derivation and the sample size of each group (*n* = 6). The control group was defined as group 1, and the experimental group was defined as group 2. The significance level of alpha was defined as 0.05.

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
