# Peer review of "Free Myocutaneous Flap Assessment in a Rat Model: Verification of a Wireless Bioelectrical Impedance Assessment (BIA) System for Vascular Compromise Following Microsurgery"

_jpm, 2021, doi:10.3390/jpm11050373_

Round 1

Reviewer 1 Report

Thank you very much for the opportunity to review this manuscript. In this manuscript, the authors address a significant topic in plastic and reconstructive surgery: Free flap monitoring. The authors evaluate the utility of a bioelectrical impedence assessment (BIA) system in a rodent model for its usefulness in monitoring the vascularity and tissue status of grafted flaps. This reviewer believes the following comments need to be addressed prior to recommending this manuscript for publication:

1- Please include a power analysis

2- In the discussion, please elaborate on how close this system is to being implemented in patients

3- Have the authors correlated their findings with metabolic parameters? (lactate, ck, mb, etc...)

4- Please comment on the cost of using such a system

5- It might be helpful to perform a comparison to other systems available for flap monitoring (oxymetry, etc...)

Thanks again for the opportunity to review this manuscript.

Author Response

Journal: JPM (journal of personalized medicine)

Manuscript ID: jpm-1190135

Title: Free myocutaneous flap assessment in a rat model: verification of a wireless bioelectrical impedance assessment (BIA) system for vascular compromise following microsurgery

Dear Editors, Reviewers and Ms. Christa Sas

We are submitting our manuscript entitled “Free myocutaneous flap assessment in a rat model: verification of a wireless bioelectrical impedance assessment (BIA) system for vascular compromise following microsurgery " for consideration of “Journal of Personalized Medicine” after revise. Thank you very much again for granting the privilege to us to revise the paper. We have specifically responded to the reviewers’ questions and criticisms point-by-point as follows and add them into this version. Any changes in the manuscript can be tracked by the tool of the MS Word and be marked by underline or colored highlight.

Reviewer 1

[Comment 1] Please include a power analysis

[Answer 1]

We add a new table 1 for this purpose at statistic section.

[Comment 2]

In the discussion, please elaborate on how close this system is to being implemented in patients

[Answer 2]

We add those description in the discussion.

“It is potentially useful in a clinical setting after microsurgery with free flaps. A firm adaptor fixing the BIA probe on various flaps surface will be the next step to clinical applications.”

[Comment 3]

Have the authors correlated their findings with metabolic parameters? (lactate, ck, mb, etc...)

[Answer 3]

Due to the subjects are tiny (small-sized rat), the blood sample is not sufficient for detailed metabolic analysis in this study.

[Comment 4]

Please comment on the cost of using such a system.

[Answer 4]

The cost of this system is about 200 U.S.D.

We add “The cost of this system is about 200 U.S.D.” in the Material and Methods.

[Comment 4]

It might be helpful to perform a comparison to other systems available for flap monitoring (oxymetry, etc...)

[Answer 4]

Thanks for your constructive suggestion. We had tried to measure saturation at the beginning of this study. However, switching between BIA system and commercialized oximeter is difficult for stable laboratory data. Thus, we gave up for compare the oximeter and bioimpedance in this study. One instrument with those two functions (BIA and oximeter ) may be necessary for this purpose.

This table is different attempt as the StO2 measurement. If you consider this separated figure helpful for the audient, we will like to attach this figure into the supplement data. 

Thanks again for your informative reviewing and the opportunity to revise this manuscript.

Reviewer 2 Report

1.The introduction needs to be improved with regards to the state of the art and the background. Tissue viability in free flaps is primarily based of proper vascularization and oxygen delivery to the tissue. Insufficient oxygen delivery will lead to tissue ischemia and anaerobic metabolism etc, that will lead to changes in glucose, lactate, pH, Na, Ka and fluid shifts between extra and intra cellular spaces and thus tissue oedema. This is essentially what will be detected with impedance. You need to discuss and describe the process to let the reader understand what is happening to the tissue and how it can be detected. StO2 sensor for example such as this (https://ieeexplore.ieee.org/abstract/document/8110619) have been used for free flap tissue viability monitoring. This example is very small in size and power consumption and complexity. You should discuss this and other similar works and compare them to your proposed system. Bioimpedance has also been used extensively for ischemia detection. For example:

https://ieeexplore.ieee.org/abstract/document/983458

https://link.springer.com/article/10.1007/s10544-006-6381-y

https://www.sciencedirect.com/science/article/pii/S0956566303002045

https://ieeexplore.ieee.org/abstract/document/7299403

As well electrochemical sensors: https://www.sciencedirect.com/science/article/pii/S0956566314001705

You need to discuss the state of the art in terms of ischemia sensors and sensing technology to provide a holistic overview of the state of the art.

2. Please provide detailed information about the developed electronic instrumentation. Components used etc are needed. How do you control the current and how do you ensure it is below 5 uA? Bioimpedance systems typically use a voltage controlled current source, such as a howland current source to inject a known current into the tissue, primarily for safety purposes but also to eliminate the need of measuring the injected current into the tissue. In addition safety limits according to internatgional standards have been set with regards to currents into tissues: 100 μA from 0.1 Hz to 1 kHz, 100f μA from 1 to 100 kHz, and 10 mA for frequencies greater than 100 kHz, where f is the frequency in kHz. a 5 uA can be beneficial in terms of power consumption but maybe not that great for SNR. Please comment of these issues in the paper. Please provide a detailed schematic of the circuitry. Why is there a need for a voltage divider? How do you measure the current in the tissue?

3. Bioimpedance is pressure sensitive. With the electrodes attached to a bulky device such as the one shown in Fig. 1 applying a constant pressure will be challenging and will lead to deviations in measured spectra. Please comment on that.

4. How has the spacing between electrodes been chosen? Electrode separation defines the depth within the tissue that will be interrogated. Please comment on that. These needs to be some electric field simulations to support this. Why have stainless steel electrodes been used? and why the particular electrode shape?

5. What is the reason for using bipolar impedance measurements? Bipolar measurement also measure the electrode interface impedance which will be the main contributor to the measured impedance. Depending on the electrodes used, 20 kHz can still be a low frequency where electrode impedance may dominate. Please also provide measurement of the electrode impedance, using 3-electrode electrochemical measurements in PBS. Consequently tetrapolar impedance measurements are preferred.

6. Ischemia measurement are usually reported up to a frequency of 1 MHz, to allow interrogation of both the intracellular and extracellular spaces. Why are your measurements limited to 20 kHz? Is there a reason for that?

7. A diagram illustrating how the each animal and their two limbs were treated would help visualize the experimental protocol and allow the reader better understand what you have done.

8."Interestingly, the bio-impedance decreased as the frequency increased. This is because higher frequency currents have a better penetrating ability in tissue": This is well known.

9.Most of the discussions in lines 179-191 with regards to the state of the art and other methods should be largely moved to the introduction.

10. "1k Hz detected the latest bio-impedance difference between the experimental group and the control group after vascular compromise.": How was vascular compromise detected and assessed?

Author Response

Journal: JPM (journal of personalized medicine)

Manuscript ID: jpm-1190135

Title: Free myocutaneous flap assessment in a rat model: verification of a wireless bioelectrical impedance assessment (BIA) system for vascular compromise following microsurgery

Dear Editors, Reviewers and Ms. Christa Sas

We are submitting our manuscript entitled “Free myocutaneous flap assessment in a rat model: verification of a wireless bioelectrical impedance assessment (BIA) system for vascular compromise following microsurgery " for consideration of “Journal of Personalized Medicine” after revise. Thank you very much again for granting the privilege to us to revise the paper. We have specifically responded to the reviewers’ questions and criticisms point-by-point as follows and add them into this version. Any changes in the manuscript can be tracked by the tool of the MS Word and be marked by underline or colored highlight.

Reviewer 2

[Comment 1]

Comments and Suggestions for Authors

1.The introduction needs to be improved with regards to the state of the art and the background. Tissue viability in free flaps is primarily based of proper vascularization and oxygen delivery to the tissue. Insufficient oxygen delivery will lead to tissue ischemia and anaerobic metabolism etc, that will lead to changes in glucose, lactate, pH, Na, Ka and fluid shifts between extra and intra cellular spaces and thus tissue oedema. This is essentially what will be detected with impedance. You need to discuss and describe the process to let the reader understand what is happening to the tissue and how it can be detected. StO2 sensor for example such as this (https://ieeexplore.ieee.org/abstract/document/8110619) have been used for free flap tissue viability monitoring. This example is very small in size and power consumption and complexity. You should discuss this and other similar works and compare them to your proposed system. Bioimpedance has also been used extensively for ischemia detection. For example:

https://ieeexplore.ieee.org/abstract/document/983458

https://link.springer.com/article/10.1007/s10544-006-6381-y

A SiC microdevice for the minimally invasive monitoring of ischemia in living tissues

https://www.sciencedirect.com/science/article/pii/S0956566303002045

Minimally invasive silicon probe for electrical impedance measurements in small animals in 2003

https://ieeexplore.ieee.org/abstract/document/7299403

A tetrapolar bio-impedance sensing system for gastrointestinal tract monitoring

As well electrochemical sensors:

in vivo ischemia monitoring array for endoscopic surgery

2014

https://doi.org/10.1016/j.bios.2014.02.080

You need to discuss the state of the art in terms of ischemia sensors and sensing technology to provide a holistic overview of the state of the art.

[Answer 1]

Sincerely thanks for your elegant comments. We updated those relevant documents with brief introduction. Those articles also citated for further reference of the readers.

[Comment 2].

Please provide detailed information about the developed electronic instrumentation. Components used etc are needed. How do you control the current and how do you ensure it is below 5 uA? Bioimpedance systems typically use a voltage controlled current source, such as a howland current source to inject a known current into the tissue, primarily for safety purposes but also to eliminate the need of measuring the injected current into the tissue. In addition safety limits according to internatgional standards have been set with regards to currents into tissues: 100 μA from 0.1 Hz to 1 kHz, 100f μA from 1 to 100 kHz, and 10 mA for frequencies greater than 100 kHz, where f is the frequency in kHz. a 5 uA can be beneficial in terms of power consumption but maybe not that great for SNR. Please comment of these issues in the paper. Please provide a detailed schematic of the circuitry. Why is there a need for a voltage divider? How do you measure the current in the tissue?

[Answer 2] Thanks for your professional and instructive comments. The design of our bio-impedance parameter monitoring device is based on the steady voltage driven excitation mode, and its structure is similar to the system in the followings (https://www.analog.com/en/technical-articles/bioelectrical-impedance-analysis-in-monitoring-of-the-clinical-status-and-diagnosis-of-diseases.html). According to the reviewer’s comments, the schematic of the designed device has been modified in the revised manuscript. Here, the excitation buffer can provide 2.5 Vp-p steady excitation voltage, and the reference resistance is set to about 5 M ohm. Therefore, the injected current will be limited below 5 uA.

[Comment 3]

Bioimpedance is pressure sensitive. With the electrodes attached to a bulky device such as the one shown in Fig. 1 applying a constant pressure will be challenging and will lead to deviations in measured spectra. Please comment on that.

[Answer 3]

We use self-assembling holder to stabilize the probe on the target surface.

[Comment 4]

How has the spacing between electrodes been chosen? Electrode separation defines the depth within the tissue that will be interrogated. Please comment on that. These needs to be some electric field simulations to support this. Why have stainless steel electrodes been used? and why the particular electrode shape?

[Answer 4]

Many thanks for the reviewer’s comments. The distance between two electrodes is set to 10 mm. Because the thickness of the flaps is about 1~3 mm, the measuring depth of the designed device should be sufficient. Here, the shape of the used electrodes is round tip needle, and it can reduce the risk of injuring the flaps. The used electrodes are made of stainless steel can improve the issue of metal oxidation.

[Comment 5]

What is the reason for using bipolar impedance measurements? Bipolar measurement also measures the electrode interface impedance which will be the main contributor to the measured impedance. Depending on the electrodes used, 20 kHz can still be a low frequency where electrode impedance may dominate. Please also provide measurement of the electrode impedance, using 3-electrode electrochemical measurements in PBS. Consequently tetrapolar impedance measurements are preferred.

[Answer 5]

Many thanks for your comments. Under the consideration of simple operation and implementation of the designed device, bipolar measurement is used in this study. According to the reviewer’s mention, in bipolar measurement, the electrode interface impedance may cause the contributor in the measured impedance. Under the same measurement condition, we focus on monitoring the relative impedance change caused from the change in the health of the flap. 

[Comment 6]

Ischemia measurement are usually reported up to a frequency of 1 MHz, to allow interrogation of both the intracellular and extracellular spaces. Why are your measurements limited to 20 kHz? Is there a reason for that?

[Answer 6]:

Many thanks for the reviewer’s comments. Bioimpedance in the alpha domain may be associated with information on tissue interfaces, while the beta domain is associated with the polarization of cellular membranes, proteins and other organic macromolecules. Therefore, we consider that the change in the health of the flap could be effectively observed by the bioimpedance measurement in the alpha and beta domain. 

Comment 7.

A diagram illustrating how each animal and their two limbs were treated would help visualize the experimental protocol and allow the reader better understand what you have done.

[Answer 7]

Thanks for your comment. We add a new figure for this purpose.

Comment 8."Interestingly, the bio-impedance decreased as the frequency increased. This is because higher frequency currents have a better penetrating ability in tissue": This is well known.

[Answer 8] We revised this paragraph. “The bio-impedance decreased as the frequency increased. This is because higher frequency currents have a better penetrating ability in tissue.”

[Comment 9]. Most of the discussions in lines 179-191 with regards to the state of the art and other methods should be largely moved to the introduction.

[Answer 9]

Thanks for your comment. We move some descriptions of the clinical manner from the discussion to the introduction. The discussions were focus on the updated technique with our BIA system.

[Comment 10].

"1k Hz detected the latest bio-impedance difference between the experimental group and the control group after vascular compromise.": How was vascular compromise detected and assessed?

[Answer 10]: Many thanks for your comments. In the animal experiment, the left thigh of the animal was surgically dissected, and its muscles, nerves and skin tissue were all divided; their vascular pedicle was isolated but not transected. The right thigh of the animal that did not undergo surgery was still able to supply blood and nutrition from the femoral vessels and was used as the control group.

Thanks again for your informative reviewing and the opportunity to revise this manuscript.

Round 2

Reviewer 1 Report

The authors have successfully addressed the comments raised by the reviewers

Author Response

Thanks for your kind review.

Reviewer 2 Report

Most of my comments have been partially addressed in the response letter but have not been addressed within the main document. Please address my previous comments with changes in the main document. Addressing them only in the response letter is not sufficient.

Please provide the name and values of the components used for your circuit.

Author Response

March 23, 2021

Journal: JPM (journal of personalized medicine)

Manuscript ID: jpm-1190135-R1

Title: Free myocutaneous flap assessment in a rat model: verification of a wireless bioelectrical impedance assessment (BIA) system for vascular compromise following microsurgery

Dear Editors, Reviewers and Ms. Christa Sas

We are submitting our manuscript entitled “Free myocutaneous flap assessment in a rat model: verification of a wireless bioelectrical impedance assessment (BIA) system for vascular compromise following microsurgery " for consideration of “Journal of Personalized Medicine” after revise. Thank you very much again for granting the privilege to us to revise the paper. We have specifically responded to the reviewers’ questions and criticisms point-by-point as follows and add them into this version. Any changes in the manuscript can be tracked by the tool of the MS Word and be marked by underline or colored highlight.

Reviewer 2

[Comment 1]

Comments and Suggestions for Authors

1.The introduction needs to be improved with regards to the state of the art and the background. Tissue viability in free flaps is primarily based of proper vascularization and oxygen delivery to the tissue. Insufficient oxygen delivery will lead to tissue ischemia and anaerobic metabolism etc, that will lead to changes in glucose, lactate, pH, Na, Ka and fluid shifts between extra and intra cellular spaces and thus tissue oedema. This is essentially what will be detected with impedance. You need to discuss and describe the process to let the reader understand what is happening to the tissue and how it can be detected. StO2 sensor for example such as this (https://ieeexplore.ieee.org/abstract/document/8110619) have been used for free flap tissue viability monitoring. This example is very small in size and power consumption and complexity. You should discuss this and other similar works and compare them to your proposed system. Bioimpedance has also been used extensively for ischemia detection. For example:

https://ieeexplore.ieee.org/abstract/document/983458

https://link.springer.com/article/10.1007/s10544-006-6381-y

A SiC microdevice for the minimally invasive monitoring of ischemia in living tissues

https://www.sciencedirect.com/science/article/pii/S0956566303002045

Minimally invasive silicon probe for electrical impedance measurements in small animals in 2003

https://ieeexplore.ieee.org/abstract/document/7299403

A tetrapolar bio-impedance sensing system for gastrointestinal tract monitoring

As well electrochemical sensors:

in vivo ischemia monitoring array for endoscopic surgery

2014

https://doi.org/10.1016/j.bios.2014.02.080

You need to discuss the state of the art in terms of ischemia sensors and sensing technology to provide a holistic overview of the state of the art.

[Answer 1]

Sincerely thanks for your elegant comments. We updated those relevant documents with brief introduction. Those articles also citated for further reference of the readers.

[Change for Comment 1]

Newly added in introduction, line 35-41

“Measuring the surface temperature of grafted skin flaps is the simplest way of monitoring skin flap grafting. Physicians used skin temperature indicators to conduct free flap monitoring[7]. When the center location of the skin dropped by more than 3ºC from the baseline, it might have encountered arterial thrombosis. However, the sensitivity for monitoring the surface temperature of skin flap grafts is not high enough[8].”

and list of new References from your comment

  1. Berthelot, M.; Yang, G.Z.; Lo, B. A Self-Calibrated Tissue Viability Sensor for Free Flap Monitoring. IEEE J Biomed Health Inform 2018, 22, 5-14, doi:10.1109/JBHI.2017.2773998.
  2. Wtorek, J.; Jozefiak, L.; Polinski, A.; Siebert, J. An averaging two-electrode probe for monitoring changes in myocardial conductivity evoked by ischemia. IEEE Trans Biomed Eng 2002, 49, 240-246, doi:10.1109/10.983458.
  3. Gomez, R.; Ivorra, A.; Villa, R.; Godignon, P.; Millan, J.; Erill, I.; Sola, A.; Hotter, G.; Palacios, L. A SiC microdevice for the minimally invasive monitoring of ischemia in living tissues. Biomed Microdevices 2006, 8, 43-49, doi:10.1007/s10544-006-6381-y.
  4. Tahirbegi, I.B.; Mir, M.; Schostek, S.; Schurr, M.; Samitier, J. in vivo ischemia monitoring array for endoscopic surgery. Biosens Bioelectron 2014, 61, 124-130, doi:10.1016/j.bios.2014.02.080.

[Comment 2].

Please provide detailed information about the developed electronic instrumentation. Components used etc are needed. How do you control the current and how do you ensure it is below 5 uA? Bioimpedance systems typically use a voltage controlled current source, such as a howland current source to inject a known current into the tissue, primarily for safety purposes but also to eliminate the need of measuring the injected current into the tissue. In addition safety limits according to internatgional standards have been set with regards to currents into tissues: 100 μA from 0.1 Hz to 1 kHz, 100f μA from 1 to 100 kHz, and 10 mA for frequencies greater than 100 kHz, where f is the frequency in kHz. a 5 uA can be beneficial in terms of power consumption but maybe not that great for SNR. Please comment of these issues in the paper. Please provide a detailed schematic of the circuitry. Why is there a need for a voltage divider? How do you measure the current in the tissue?

[Answer 2] Thanks for your professional and instructive comments. The design of our bio-impedance parameter monitoring device is based on the steady voltage driven excitation mode, and its structure is similar to the system in the followings (https://www.analog.com/en/technical-articles/bioelectrical-impedance-analysis-in-monitoring-of-the-clinical-status-and-diagnosis-of-diseases.html). According to the reviewer’s comments, the schematic of the designed device has been modified in the revised manuscript. Here, the excitation buffer can provide 2.5 Vp-p steady excitation voltage, and the reference resistance is set to about 5 M ohm. Therefore, the injected current will be limited below 5 uA.

[Change for Comment 2]

Line 87-93

“Bioimpedance systems typically use a voltage controlled current source, such as a Howland current source to inject a known current into the tissue, primarily for safety purposes but also to eliminate the need of measuring the injected current into the tissue. In addition, safety limits according to internatgional standards have been set with regards to currents into tissues: 100 μA from 0.1 Hz to 1 kHz, 100f μA from 1 to 100 kHz, and 10 mA for frequencies greater than 100 kHz, where f is the frequency in kHz. A 5 μA can be beneficial in terms of power consumption but maybe not that great for signal-to-noise ratio.”

Line 99-106

“The excitation buffer can provide 2.5 Vp-p steady excitation voltage, and the reference resistance is set to about 5 M ohm. Therefore, the injected current will be limited below 5 uA.”

We add a new detailed schematic of the circuitry as Figure 1

Figure 1. (a) Photograph and (b) block diagram of the self-assembled bio-impedance parameter monitoring device.

[Comment 3]

Bioimpedance is pressure sensitive. With the electrodes attached to a bulky device such as the one shown in Fig. 1 applying a constant pressure will be challenging and will lead to deviations in measured spectra. Please comment on that.

[Answer 3]

We use self-assembling holder to stabilize the probe on the target surface.

[Change for Answer 3]

line 141

“The BIA system probes were placed on the thighs of the animal to acquire the bio-impedance every hour using self-assembling holder to stabilize the probe on the target surface.”

[Comment 4]

How has the spacing between electrodes been chosen? Electrode separation defines the depth within the tissue that will be interrogated. Please comment on that. These needs to be some electric field simulations to support this. Why have stainless steel electrodes been used? and why the particular electrode shape?

[Answer 4]

Many thanks for the reviewer’s comments. The distance between two electrodes is set to 10 mm. Because the thickness of the flaps is about 1~3 mm, the measuring depth of the designed device should be sufficient. Here, the shape of the used electrodes is round tip needle, and it can reduce the risk of injuring the flaps. The used electrodes are made of stainless steel can improve the issue of metal oxidation.

[Change for Answer 4]

Line 121 to 125

“The distance between two electrodes is set to 10 mm. Because the thickness of the flaps is about 1~3 mm, the measuring depth of the designed device should be sufficient. Here, the shape of the used electrodes is round tip needle, and it can reduce the risk of injuring the flaps. The used electrodes are made of stainless steel can improve the issue of metal oxidation.”

[Comment 5]

What is the reason for using bipolar impedance measurements? Bipolar measurement also measures the electrode interface impedance which will be the main contributor to the measured impedance. Depending on the electrodes used, 20 kHz can still be a low frequency where electrode impedance may dominate. Please also provide measurement of the electrode impedance, using 3-electrode electrochemical measurements in PBS. Consequently tetrapolar impedance measurements are preferred.

[Answer 5]

Many thanks for your comments. Under the consideration of simple operation and implementation of the designed device, bipolar measurement is used in this study. According to the reviewer’s mention, in bipolar measurement, the electrode interface impedance may cause the contributor in the measured impedance. Under the same measurement condition, we focus on monitoring the relative impedance change caused from the change in the health of the flap. 

[Change for comment 5]

None

[Comment 6]

Ischemia measurement are usually reported up to a frequency of 1 MHz, to allow interrogation of both the intracellular and extracellular spaces. Why are your measurements limited to 20 kHz? Is there a reason for that?

[Answer 6]:

Many thanks for the reviewer’s comments. Bioimpedance in the alpha domain may be associated with information on tissue interfaces, while the beta domain is associated with the polarization of cellular membranes, proteins and other organic macromolecules. Therefore, we consider that the change in the health of the flap could be effectively observed by the bioimpedance measurement in the alpha and beta domain. 

[Change for comment 6]

None

Comment 7.

A diagram illustrating how each animal and their two limbs were treated would help visualize the experimental protocol and allow the reader better understand what you have done.

[Answer 7]

Thanks for your comment. We add a new figure for this purpose.

Change for Comment 7

We add a figure 2 for better comprehensiveness.

Comment 8."Interestingly, the bio-impedance decreased as the frequency increased. This is because higher frequency currents have a better penetrating ability in tissue": This is well known.

[Answer 8] We revised this paragraph.

[Change for comment 8]

“The bio-impedance decreased as the frequency increased. This is because higher frequency currents have a better penetrating ability in tissue.”

[Comment 9]. Most of the discussions in lines 179-191 with regards to the state of the art and other methods should be largely moved to the introduction.

[Answer 9]

Thanks for your comment. We move some descriptions of the clinical manner from the discussion to the introduction. The discussions were focus on the updated technique with our BIA system.

[Change for Comment 9]

We rearrange descriptions of older manner to the introduction. The discussion focusses on the comparison among newer devices with our BIA system.

Delete “Measuring the surface temperature of grafted skin flaps is the simplest way of monitoring skin flap grafting. Physicians used skin temperature indicators to conduct free flap monitoring. When the center location of the skin dropped by more than 3ºC from the baseline, it might have encountered arterial thrombosis. However, the sensitivity for monitoring the surface temperature of skin flap grafts is not high enough”

[Comment 10].

"1k Hz detected the latest bio-impedance difference between the experimental group and the control group after vascular compromise.": How was vascular compromise detected and assessed?

[Answer 10]: Many thanks for your comments. In the animal experiment, the left thigh of the animal was surgically dissected, and its muscles, nerves and skin tissue were all divided; their vascular pedicle was isolated but not transected. The right thigh of the animal that did not undergo surgery was still able to supply blood and nutrition from the femoral vessels and was used as the control group.

[Change for comment 10]

None.

[Comment 11 in revise 2] 

Please provide the name and values of the components used for your circuit.

[Answer 11]

We add this information in the supplements

Component

ID

Manufactory

Micro Controller Unit

RX210

Renesas

Bluetooth

RN4678

Microchip

Instrumentation Amplifiers

LT1789

Analog Devices

Voltage Regulator

LP3985

Texas Instruments

Operational Amplifiers

AD8607

Analog Devices

Operational Amplifiers

LMP7704

Texas Instruments

Thanks again for your informative reviewing and the opportunity to revise this manuscript.

Sincerely,

Bor-Shyh Lin,PhD.

Institute of Imaging and Biomedical Photonics,

National Chiao-Tung University, Tainan, Taiwan

E-mail: borshyhlin@gmail.com

Round 3

Reviewer 2 Report

  1. You response to comments 5 and 6 should be addressed in the main document not just in your response letter. Please include the reasoning behind choosing bipolar measurements and measurements at one low frequency value in the main document.
  2. The component list that you have provided as a  supplementary can be incorporated into the circuit schematic. Please do that. Also please provide additional circuit details, such as voltage measurement amplification gain. What are the values of all the passive you used in your circuit, especially the reference resistance that sets the current? What is the cut-off frequency of the low-pass filter for the signal excitation channel? How was this low pass filter implemented. What was the resolution and speed in the digitization of the analog signal. Phase delays are important for accurate calculation of impedance. Are these an issue with your instrumentation and measurement at 20 kHz? How confident are you that the current stays at the required set value and is not influenced by changes in the load? This could be an issue due to the high electrode impedance and its variations. How is the impedance computed once the analogue signal is digitized ad transmitted? Do you use any additional digital filters? Are you performing an FFT? Synchronous demodulation? All these information are missing and must be included in the main manuscript and definitely not just in the response letter. The requested changes are minor.

Author Response

Reviewer 2 Round 3

[Comment 1]

Your response to comments 5 and 6 should be addressed in the main document not just in your response letter. Please include the reasoning behind choosing bipolar measurements and measurements at one low frequency value in the main document.

[Answer to comment 1]

Many thanks for the reviewer’s comments. The mentioned information has been added in the revised manuscript.

[Change in manuscript]

line 78 to line 87

Multiple-frequency BIA can be easily classified into several different frequency bands: the alpha domain (1k Hz ~ 10 k Hz), the beta domain (10k Hz ~ 50 k Hz) and the gamma domain (>100 k Hz). Bio-impedance in the alpha domain may be associated with information on tissue interfaces, while the beta domain is associated with the polarization of cellular membranes, proteins and other organic macromolecules. Bio-impedance in the gamma domain may be associated with the polarization of water molecules. Bio-impedance in the alpha and beta domains are most frequently used because the differences between normal and pathological tissues can be observed in their variations. Therefore, the change in the health of the flap may be effectively observed by the bioimpedance measurement in the alpha and beta domain.

[Comment 2]

The component list that you have provided as a supplementary can be incorporated into the circuit schematic. Please do that. Also please provide additional circuit details, such as voltage measurement amplification gain. What are the values of all the passive you used in your circuit, especially the reference resistance that sets the current? What is the cut-off frequency of the low-pass filter for the signal excitation channel? How was this low pass filter implemented. What was the resolution and speed in the digitization of the analog signal. Phase delays are important for accurate calculation of impedance. Are these an issue with your instrumentation and measurement at 20 kHz? How confident are you that the current stays at the required set value and is not influenced by changes in the load? This could be an issue due to the high electrode impedance and its variations. How is the impedance computed once the analogue signal is digitized ad transmitted? Do you use any additional digital filters? Are you performing an FFT? Synchronous demodulation? All these information are missing and must be included in the main manuscript and definitely not just in the response letter. The requested changes are minor.

[Answer to comment 2]

Many thanks for the reviewer’s comments. The gain of the voltage acquisition circuit is adjustable, and can be set to 1, 2, 5, or 10. The cut-off frequency of the low-pass filter in the steady voltage source circuit is set to 25k Hz. The sampling rate of the analog-to-digital converter in the microprocessor is set to 200k Hz. The above information has been added in the revised manuscript. The design of the used bio-impedance parameter monitoring device is developed in cooperation with other companies. Therefore, it is not suitable to release the detailed circuits due to the issue of intellectual property.

[Change in manuscript]

line 117 to line 129

The cut-off frequency of the low-pass filter in the steady voltage source circuit was set to 25k Hz. The generated steady voltage will then pass through the voltage divider circuit and the stainless-steel electrodes. The pair of stainless-steel electrodes will be placed on the region of tissue being measured. When generated, the steady voltage passes through the tissue via the electrodes, and will then be attenuated due to tissue bio-impedance. The voltage acquisition circuit then receives the attenuated steady voltage signal and estimates the bio-impedance of the tissue. The gain of the voltage acquisition circuit is adjustable, and can be set to 1, 2, 5, or 10. The sampling rate of the analog-to-digital converter in the microprocessor is set to 200k Hz. Under the consideration of simple operation and implementation of the designed device, bipolar measurement is used in this study. In bipolar measurement, the electrode interface impedance may cause the contributor in the measured impedance. Under the same measurement condition, the relative impedance change caused from the change in the health of the flap was monitored.